# Blood Service in a Region of China’s Qinghai–Tibetan Plateau

**DOI:** 10.3390/healthcare11131944

**Published:** 2023-07-05

**Authors:** Pan Sun, Liyuan Zhu, Li Ma, Changqing Li, Zongkui Wang, Rong Zhang, Shengliang Ye, Ya Wang

**Affiliations:** 1Institute of Blood Transfusion, Chinese Academy of Medical Sciences and Peking Union Medical College, Chengdu 610052, China; pan.sun@ibt.pumc.edu.cn (P.S.); sxszhuliyuan@student.pumc.edu.cn (L.Z.); mary19820225@163.com (L.M.); lichangqing268@163.com (C.L.); zongkui.wang@ibt.pumc.edu.cn (Z.W.); rong.zhang@ibt.pumc.edu.cn (R.Z.); 2Blood Center of Aba Tibetan and Qiang Autonomous Prefecture, Barkam 624000, China

**Keywords:** blood service, China, ethnic minority, Qinghai–Tibetan Plateau

## Abstract

**Objectives:** The purpose of this paper is to describe blood services in the Aba Tibetan and Qiang Regions, (hereinafter referred to as Aba Prefecture), a region of China’s Qinghai–Tibetan Plateau, the third largest area of Tibet and the main inhabited area of the Qiang people. **Design:** We present a comprehensive investigation into blood donations, donors, screening and supply in the 13 counties of Aba Prefecture based on data from 2013 to 2018. Geography and population were also used to analyze the differences in blood services among different regions. **Participants:** The number of blood donors totaled 19,047. **Results:** Over the past 6 years, blood donations have increased by 29 and clinical blood usage by 45%. The blood donation rate was 3.4‰ and per capita blood use was 1.04 mL, both of which were significantly lower than the national average, and blood donation decreased with altitude. It should be noted that the donation rate of the Tibetan and Qiang peoples is much lower than that of the Han population. Moreover, the rejection rate of blood in laboratory testing was found to be higher than the national average, especially in counties located at higher altitudes. **Conclusions:** Blood donations and usage increased every year in Aba Prefecture, but blood shortage is still an important issue. In addition, the prevalence of transfusion–transmitted diseases is relatively high, which may be linked to lower-education and unfavorable geographical and medical conditions.

## 1. Introduction

The Aba Tibetan and Qiang regions, commonly known as Aba Prefecture, is a region in Sichuan province, on the southeastern edge of the Qinghai–Tibetan Plateau. Aba Prefecture is the third-largest Tibetan area and the main inhabited area for China’s Qiang people. It covers 13 counties covering an area of 84,200 square kilometers with a population of 940,000. The region is predominantly characterized by plateaus and mountain valleys. The average altitude is 3500–4100 m, and the altitude of the county seats (the major population centers) varies from 1326 to 3406 m [1]. The population of Aba Prefecture comprises many ethnic groups, of which the Tibetan, Qiang, and Hui account for 58.7, 18.5, and 3.2%, respectively, while the Han Chinese account for 19.4%. Farmers and herdsmen constitute 78.1% of the total population [1,2]. A local blood center serves both residents and tourists, and although donations have been previously investigated [3], systematic and comprehensive blood services, including donations, screening and supply in the ethnic Tibetan and Qiang areas has rarely been reported in detail. To gain a more comprehensive understanding of blood services in the ethnic minority regions of China, a descriptive study of the geography, population and blood services––including voluntary non-remunerated blood donation, donors, screening and supply––in Aba Prefecture was conducted from 2013 to 2018. This investigation is anticipated to furnish valuable insights into the development of a secure and sufficient blood supply.

## 2. Methods

### 2.1. Study Setting and Design

The data of blood donation, donor testing and supply in the 13 counties were collected by the Aba Prefecture blood center. Geographical and population data for each county in 2018 were obtained from the people’s government of Aba Prefecture. All data were entered and encoded using Microsoft Excel and analyzed using IBM SPSS 25.0 software (IBM Corporation, NY, USA). Descriptive statistics were displayed, and the sociodemographic characteristics of blood donors, (such as age, ethnicity, profession and education) were analyzed. Additionally, multi-group comparisons for blood donation and blood use among the different counties were performed using one-way ANOVA, followed by Tamhane’s T2 comparison test as the heterogeneity of variance [4]. A Pearson correlation was used to analyze the correlation between the blood donation rate and the altitude of the county. The confidence interval was defined as 95%, and a *p*-value of less than 0.05 was considered statistically significant.

### 2.2. Patient and Public Involvement

There was no direct patient or public involvement in this study, and all data were collected online. Whole blood donations between 2013 and 2018 were extracted from the Aba Prefecture blood management information system.

## 3. Results

### 3.1. Blood Donation 

It should be noted that all donations (whole blood with no apheresis components) were collected by the Aba Prefecture central blood center. Furthermore, the majority of donations were collected on mobile donation vehicles on the street. Data on geography, population, blood donation and blood use of the 13 counties are presented in Table 1. Over the past six years, a total of 19,047 blood donations were made, of which 3554 were made in 2018, representing a notable increase of 29.04% over the 2704 donations in 2013. Additionally, an annual increase of 1.04% was observed compared to 2017 as reported in the previous study [3]. The volume of blood collected from 2013 to 2018 also maintained a steady increase at the same rate as donations. The volume of donated blood in 2018 was 905 L, an increase of 29% from 2013. As detailed in Table 1 and Figure 1, the average annual volume of collected blood was 821.8 L. 

The number of blood donations varied across different counties. Figure 2 shows the number of donations and donation rates per 1000 in the 13 counties. A one-way ANOVA revealed a significant difference in donation rates among the various counties (*p* < 0.05). In six counties––Barkam, Wenchuan, Li, Mao, Songpan and Jiuzhaigou––the donation rate per 1000 was higher than 2.9, while the rate was lower than 2.5 in the other seven. The comparison between counties showed that the average donation rate in Barkam and Wenchuan was significantly higher. Moreover, the donation rate in Hongyuan, Rangtang and Aba, with county-seat altitude exceeding 3000 m, was significantly lower. Barkam serves as the capital of Aba Prefecture, while the only higher-education institute is situated in Wenchuan. These factors may account for the higher number of donors and greater amount of donated blood in these two countries. To estimate the relative importance of county-seat altitude on donation rate, a Pearson correlation analysis was conducted using donation data from counties other than Barkam or Wenchuan. The results indicated a significant negative correlation between the donation rates and the altitude of their respective county seats (Pearson correlation −0.461, *p* < 0.05), which implied that as the altitude of the county seats increased, the donation rates decreased.

### 3.2. Blood Donors

Figure 3 presents the demographic characteristics of the donors. The majority were between 18 and 45, with the number of donors decreasing with age. Especially in Wenchuan, more than 68% of donors belonged to the 18–25 age group (Figure 3A). The previously mentioned higher-education institute in Wenchuan played a significant role in attracting student blood donors, which contributed to a lower average donor age in that county. Han Chinese donors comprised 41.2% of the total donor population, followed by Tibetan (33.4%), Qiang (18.7%), Hui (4.7%) and 17 other ethnic groups (2%). The main ethnic blood-donor groups in each county varied due to the different distribution of populations. For example, there were more Han Chinese donors in Barkam, Wenchuan and Jiuzhaigou, while more Tibetan donors were found in Jinchuan, Xiaojin and Ruoergai. Qiang donors were mainly from Mao, and Hui donors were predominantly distributed in Aba and Songpan (Figure 3B). An analysis of the profession of the donors revealed farmers and herdsmen, students, civil servants, workers (who work in factories), staff members (including administrative staff and drivers in the companies), teachers, medical staff, military personnel and individuals from other occupations. Civil servants constituted the main group in most counties. However, the proportion of farmers and herdsmen among donors in Jiuzhaigou, Mao, Jinchuan and Xiaojin was higher. As mentioned above, many donors in Wenchuan were college students (Figure 3C). The education background of donors was primarily junior college, university or above, and senior high or technical secondary school. Donors with high education (junior college, university or above) accounted for 50.3% of the total, whereas those with junior high school or below only accounted for 19.5%. In Wenchuan, Ruoergai, Heishui and Songpan, more than 65% of donors had a higher-education qualification. Conversely, the majority of donors in Jiuzhaigou and Mao, 66.7 and 63.6%, respectively, had not received higher education (Figure 3D).

### 3.3. Blood Screening

Before donating, blood donors underwent pre-screening consisting of a medical history questionnaire, physical examination and rapid tests that measured hemoglobin concentration (Hb), hepatitis B surface antigen (HBsAg), alanine aminotransferase (ALT) and ABO blood group. A pre-screening kit for HBsAg was conducted using a rapid detection kit that produced a small but expected rate of false-negatives. After donation, laboratory tests for ABO group, HBsAg, hepatitis C virus (HCV) antibodies, ALT, Human immunodeficiency virus (HIV)-1 and HIV-2 antibodies and treponemal (TP) antibodies were conducted. Tests for transfusion-transmitted disease (TTD) used two rounds of enzyme-linked immunosorbent assays (ELISA) with two test kits from different manufacturers. If one of the tests yielded a positive result, the corresponding blood products were disqualified from transfusions. Subsequently, the samples were sent to the Chengdu Blood Center for nucleic acid testing (NAT) of hepatitis B virus (HBV), HCV and HIV. As shown in Table 2, the rejection rate of blood in laboratory tests was 4.75% (904 donors). Among the unqualified donations detected in laboratories, the majority (69.05%) were attributed to failing the ALT screening. Positive TP results were the primary reason for donor deferral in TTD tests, while HBV also accounted for a significant proportion. The results also indicated higher laboratory rejection rates in five counties––Jinchuan, Aba, Ruoergai, Hongyuan and Rangtang––closest to the center of the Qinghai–Tibet Plateau. Notably, four of these have county seats at an altitude higher than 3000 m (Table 1 and Table 2). The prevalence rate of TP was much higher in Jinchuan, Aba, Ruoergai, Hongyuan, Rangtang, while the prevalence of HBV was particularly elevated in Ruoergai and Hongyuan.

### 3.4. Blood Supply

There were almost no whole blood or platelet components used in Aba Prefecture during 2013–2018. Red blood cells and plasma were the main components, and in this study, the quantity of red blood cells represented the total quantity of supplied blood. A total of 5805.6 L of blood was supplied for clinical use by the blood center from 2013 to 2018. The 1138.8 L supplied in 2018 represented an increase of 45.07% over the 785.0 L in 2013 (Figure 1). On average, 967.6 L were used per year, which was more than the volume of blood donated (Table 1). External support from Chengdu, Mianyang and other cities in Sichuan Province was needed every year. The annual per capita blood use was 1.03 mL. Figure 4 shows the volume and per capita blood use in over the six years. The per capita blood use in the counties ranged from 0.20 mL/year in Li to 7.07 mL/year in Barkam, the capital of Aba Prefecture, which has the only tertiary-grade A-class hospital and the best medical conditions. Per capita blood use in Barkam was much higher than in the other 12 counties, among which Songpan and Jiuzhaigou had significantly higher usage.

## 4. Discussion

Maintaining a safe and adequate blood service is an important responsibility for every country [4]. China has now developed a relatively safe and complete blood service system, but in different regions it is influenced by notable differences in geography, culture, economy and demographic changes [5]. Aba Prefecture, an ethnic minority region, is located at high altitude and has economically deprived rural areas giving it a unique blood service situation. Donations increased over the past 6 years, and the annual volume of blood donations and the number of donors increased by 4.84%. The donation rate per 1000 in this area was 3.8 in 2018, higher than for Tibet (0.5), the largest Tibetan area in China [6], but far lower than for Sichuan Province (9.3) or China nationally (11.1). In the Xinjiang Uyghur Autonomous Region, another large ethnic area, the donation rate per 1000 was 6.3 [7,8]. The World Health Organization estimated that a gross donation rate of no fewer than 10–30 per 1000 was required to meet their basic clinical transfusion demands [9]. There were wide variations in donation rates among the counties. The annual whole blood donation rate (median) was 31.5 donations per 1000 (range 10.9–53.0) in high income counties [10]. Therefore, there is still a large gap between the Tibetan-inhabited area and other ethnic regions, let alone high-income counties. 

Most people in Aba Prefecture live at an altitude of 2000–4000 m in remote rural mountainous areas that have poor natural and traffic conditions. The average population density in this area is 11 people per square kilometer, far below that of Sichuan Province (171 people/km^2^) [11]. The majority of the donors were aged between 18 and 45, and the number of donors decreased with age. Thus, age may influence the proportion of donations and the rate. We did not collect the age structure for this study, a limitation that we will pay attention to in further research. Moreover, the percentage of donors of different age groups in 2018 was: 18–25 (32.7%), 26–35 (29.8%), 36–45 (25.7%) and 46–55 (11.8%), while the percentages for China nationally were, respectively, 27.5, 25.1, 24.9 and 21.2% [6]. With the economic development and social progress in ethnic areas, young people are better educated and more open-minded, which may explain why younger people accounted for a bigger proportion of blood donors in Aba Prefecture. Blood was collected mostly using collection vehicles on the street, but they were greatly affected by bad natural conditions. It is hard to conduct recruitment and blood collection in relatively bad weather and traffic conditions, so a statistical analysis suggested that the blood donation rate decreased as the altitude increased. In this study, we found that most donors were civil servants, students, and Han Chinese with higher-education. Farmers and herdsmen, who comprise 80% of the population, accounted for less than 14% of blood donors. Most Tibetan regions are remote, rural and economically deprived. Due to the different language, and lower-education, people lack awareness of voluntary blood donation and knowledge of its health benefits [12,13,14]. Moreover, the recruitment of local Tibetan residents was difficult due to their religious beliefs, where blood is regarded as a holy gift that should be well preserved [15]. Therefore, professional, public education should be strengthened to help people learn the benefits of blood donation and eventually improve the disparities with other regions. 

In 2018, the per capita blood use per year of Aba Prefecture was 1.03 mL, which is far lower than for Sichuan Province (2.8 mL) or China (3.2 mL), but higher than for Tibet (0.3 mL) [5,6,7]. This could possibly be attributed to traditional Tibetan medicine and poor medical conditions in Tibetan-inhabited areas. In addition, the medical resources in Aba Prefecture are unevenly distributed as most people receive blood transfusion treatment in Barkam. Moreover, the annual volume of blood used increased in recent years. Blood shortage is still an important issue in Aba Prefecture, and the aging population is a problem in all areas of the country, which may further widen the imbalance between blood supply and demand [16]. Governments at all levels are attaching increasing importance to minority area blood collection and supply. “Group” tailored assistance, which designates provinces or cities to provide tailored assistance to minority areas, has been conducted since 2011. Aba Prefecture now can ensure an adequate blood supply with external support, which accounts for about 27% of total use, while in Tibet it accounts for 82% [6]. 

The risk of transfusion-transmitted disease (TTD) still challenges blood services in China [17], but tests for TTD ensure the safety of the blood source. A total of 904 (4.75%) of all donations were discarded during routine laboratory tests in the past six years, higher than the average levels for China (2.40%) [6]. Among the infectious markers for transfusion-transmitted disease, disposal related to ALT screening occupied the largest proportion. Due to the high prevalence of hepatitis in China, ALT screening is specifically applied to the Chinese blood system. In the past decade, ALT screening rejected huge numbers of healthy donors [18], so ALT tests have now been largely replaced by specific hepatitis, serological and molecular tests in many counties [19]; however, an abnormal ALT level could be caused by alcohol consumption, medication or fatigue, which has no relation to hepatitis infection [20]. Therefore, the value of ALT in blood donation screening should be revisited. Positive TP and HBV results were the main reasons for blood discarding in TTD tests in Aba Prefecture. This was consistent with previous reports concerning the prevalence of TTD among donors from 2006 to 2008 [21], and 2011 to 2015 [15]. Moreover, the prevalence of TP for donors increased, at a rate higher than for HBV. This may have been related to the matriarchal tradition where unsettled and sexually unfixed marriages are still kept. Our data also demonstrated that the prevalence of TTD was much higher in donors from areas higher than 3000 m, where Tibetans, herders and religious people are the main population. In 2014, the central government financed an investment of CNY 1 billion in support of full NAT coverage at all blood centers [22]. With this support, Aba Prefecture set up a standard laboratory for NAT screening in 2018. The Ministry of Health (MOH) required nationwide implementation of NAT by the end of 2015. However, routine NAT screening still has not been carried out in Aba Prefecture. All blood samples are sent to Chengdu Blood Center for screening, which is inconvenient and uneconomical. Efforts should be made to implement NAT screening locally as soon as possible. These appropriate interventions will secure safe blood supply in this region.

Blood supply is an essential part of public health-care services, but outside help is only temporary; therefore, substantial improvement in autogenous capacity to guarantee a safe blood supply is still needed in Aba. Under the support of local government, blood collection and supply institutions hold publicity and advocacy events to enhance and encourage the public’s awareness and participation in voluntary non-remunerated blood donation. Every year, through mainstream media platforms––radio, television, newspapers, magazines––and modern communications services like QQ and WeChat, a diversity of events is launched on World Blood Donor Day, World Red Cross Day and other publicity days, as well as at festivals such as the Spring Festival. We hope to establish a safe and adequate blood service system in Aba Prefecture through the close collaboration and efforts of all relevant parties, including researchers, blood bank staff and government.

### Strengths and Limitations of This Study 

We conducted a systematic and comprehensive study on the status of blood service in the Aba Tibetan and Qiang Regions (Aba Prefecture) from 2013 to 2018 that included geography, population, voluntary non-remunerated blood donation, donors, screening and supply.The special characteristics of Aba Prefecture make the blood donation service there different from that in other part of China.The results may only reflect blood donation in Aba Tibetan and Qiang Regions, not be generalizable over other minority groups at high-altitude regions.

## 5. Conclusions

Our survey was a preliminary investigation into blood services ranging from blood donation, donors, screening and supply in Aba Prefecture, a region of China’s Qinghai-Tibetan Plateau. Further research is needed, and measures should be taken to control the transmission of epidemics and to ensure adequate and sustainable availability of blood.

## Figures and Tables

**Figure 1 healthcare-11-01944-f001:**
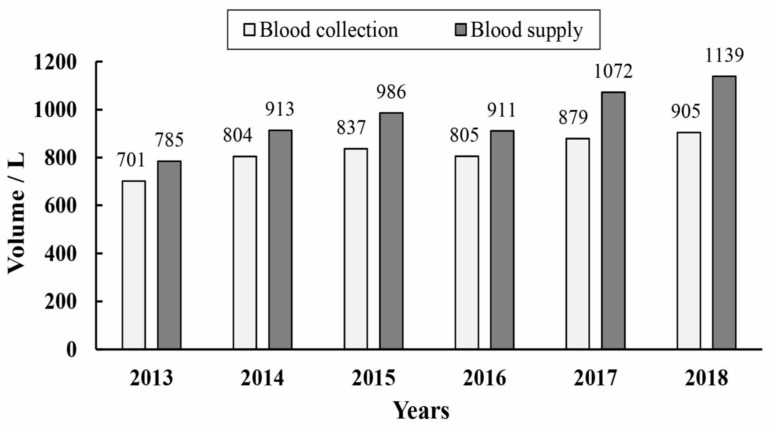
The volume of blood collection and supply in Aba Prefecture, 2013–2018.

**Figure 2 healthcare-11-01944-f002:**
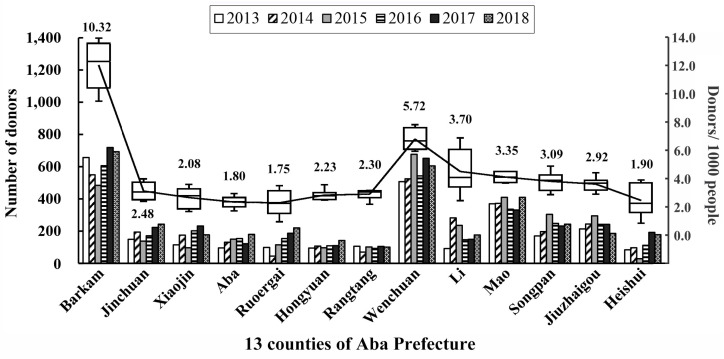
Numbers of blood donations (Histogram) and donation rates per 1000 (Box-plot) in the 13 counties of Aba Prefecture, 2013–2018.

**Figure 3 healthcare-11-01944-f003:**
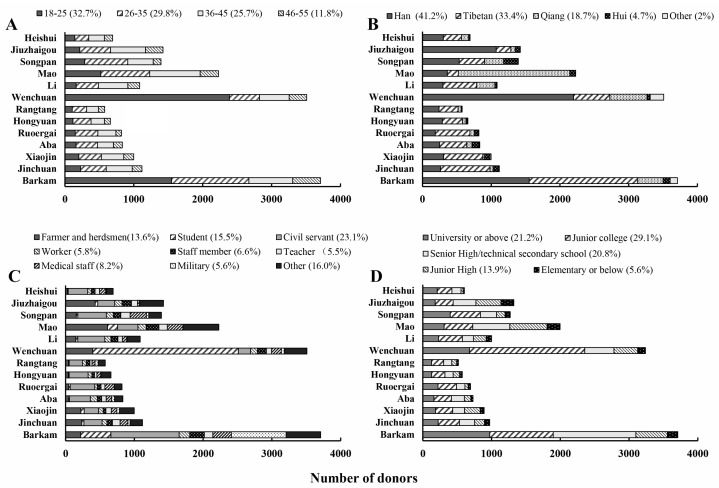
Demographic profile of blood donations collected from donors in the 13 counties of Aba Prefecture, 2013–2018. (**A**): Age group, (**B**): Ethnic group, (**C**): Profession, (**D**): Education background.

**Figure 4 healthcare-11-01944-f004:**
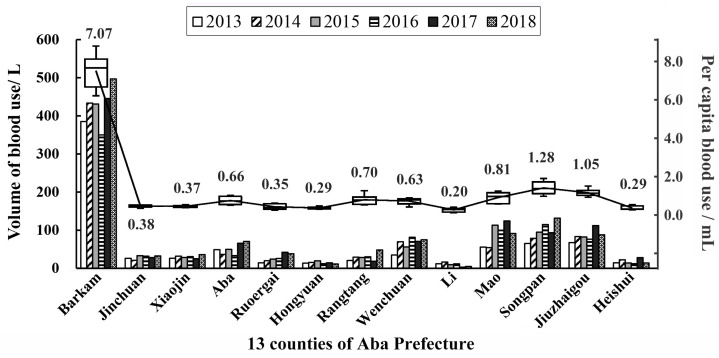
Volume of blood use (histogram) and per capita blood use (box-plot) in the 13 counties of Aba Prefecture, 2013–2018.

**Table 1 healthcare-11-01944-t001:** Geographical, population, blood donation and blood use data of the 13 counties in Aba Prefecture, 2013–2018.

Counties	Area/km^2^	Altitude /m ^a^	Population /1000 ^b^	Number of Blood Donors per Year	Volume of Blood Donation per Year /L	Volume of Blood Use per Year/L
Barkam	6639	2600	59.9	618 ± 90	159.6 ± 21.0	423.6 ± 50.8
Jinchuan	5524	2165	75.0	186 ± 41	49.1 ± 10.3	28.9 ± 4.7
Xiaojin	5571	2367	80.0	166 ± 51	43.1 ± 13.0	29.8 ± 4.0
Aba	10,435	3290	77.0	139 ± 29	36.4 ± 7.9	50.9 ± 15.0
Ruoergai	10,437	3406	78.1	137 ± 63	35.3 ± 14.8	27.5 ± 10.7
Hongyuan	8398	3504	49.3	110 ± 17	27.1 ± 4.5	14.5 ± 3.0
Rangtang	6836	3285	41.9	96 ± 14	24.4 ± 2.7	29.3 ± 10.5
Wenchuan	4083	1326	102.3	585 ± 70	140.4 ± 14.6	64.7 ± 16.7
Li	4318	1888	48.7	181 ± 68	49.8 ± 20.1	9.7 ± 4.9
Mao	4075	1580	110.8	372 ± 34	101.3 ± 10.7	89.8 ± 29.2
Songpan	8486	2851	75.2	232 ± 46	59.6 ± 14.2	96.5 ± 24.0
Jiuzhaigou	5286	1406	81.3	237 ± 36	66.3 ± 11.3	85.0 ± 15.1
Heishui	4154	2350	60.6	115 ± 61	29.3 ± 14.3	17.5 ± 6.3
Total	84,242	—	940.1	3175 ± 305	821.8 ± 71.4	967.6 ± 126.5

^a^: The altitude is that of the country seat (the major population centers); ^b^: The population is the permanent population of each county in 2017 as provided by the Statistics Bureau of Aba Prefecture. There was a minimal change in the population from 2013 to 2018, so the donation rate per 1000 was calculated using 2017 data.

**Table 2 healthcare-11-01944-t002:** Positive rates in laboratory testing.

Counties	Rejection Rates in Laboratory Testing (%) ^a^					
	ALT	HBV	HCV	TP	HIV	Total
Barkam	3.45	0.43	0.08	0.78	0.08	4.83
Jinchuan	4.11	0.45	0.27	1.07	0.27	6.17
Xiaojin	4.11	0.50	0.10	0.50	0	5.21
Aba	4.80	0.24	0.36	1.08	0	6.48
Ruoergai	4.88	0.73	0.24	1.10	0.12	7.07
Hongyuan	4.55	0.91	0.30	1.06	0.61	7.42
Rangtang	3.81	0.35	0.17	1.04	0.17	5.55
Wenchuan	2.28	0.77	0.17	0.40	0.03	3.65
Li	3.50	0.18	0.09	0.55	0	4.33
Mao	2.56	0.40	0.13	0.54	0.04	3.68
Songpan	2.58	0.79	0.14	0.43	0	3.95
Jiuzhaigou	2.95	0.63	0	0.56	0.49	4.64
Heishui	3.47	0.14	0.14	0.58	0.43	4.77
Total	3.28	0.53	0.15	0.67	0.13	4.75

^a^ After blood donation, alanine aminotransferase (ALT) and transfusion-transmitted disease (TTD) were screened. Tests for TTD (HBV, HCV, TP and HIV) used two rounds of ELISA with two test kits from different manufacturers, nucleic acid testing (NAT) for HBV, HCV and HIV were also implemented. When one of the tests was found to be positive, blood products were disqualified from transfusions.

## Data Availability

No additional data are available.

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
