# Peer review of "Blood Service in a Region of China’s Qinghai–Tibetan Plateau"

_healthcare, 2023, doi:10.3390/healthcare11131944_

Round 1
Reviewer 1 Report (Previous Reviewer 5)
All the concerns were addressed in the current edtion.
Minor editing of English language is required.
Author Response
Response 1: Thank you for this comment. We reworded the sentences (Page 1, lines 39-42).
“Although blood donation in Aba was investigated previously [3], systematic and comprehensive blood services including blood donation, screen and blood supply in Chinese Tibetan and Qiang ethnic areas was rarely reported in detail.”
2、Comment 2:Add the study type, descriptive? (Page 1, lines 43)
Response 2: We added the study type (Page 1, lines 45-46;Page 2, lines 1-3).
“To gain a more comprehensive understanding of the current status of the blood service in ethnic minority regions of China, a descriptive study was conducted to investigate the the geography, population, blood services including voluntary non-remunerated blood donation, donors, screening and supply in Aba Prefecture from 2013 to 2018.”
3、Comment 3:2.2 (Page 2, lines 14) what about ethics approval? Sample size? Add paragraph about the variables in this study. Sociodemographic characteristics of donors. how was for example education categorized?
Response 3: The data used in the study was collected from the database and online information. There was no direct patient and public involvement. Therefore the Ethics Committee waived the need for ethics approval. And no specific sample number involved. We analyzed sociodemographic characteristics of the blood donors in results section 3.2 and Figure 3, including age, ethnic, profession and education. While refer to education, five parts were divided: university or above, junior college, senior high/technical school, junior high and elementary or below. We added some description about the variables in the study setting and design (Page 2, lines 12-13).
“Descriptive statistics were displayed, and the sociodemographic characteristics of blood donors, (such as age, ethnic, profession and education background) were analyzed as well.”
4、Comment 4:4 (Page 6, lines 11) discussion: The results should be more discussed with the findings from the other studies! More critical explanation is needed for the obtained results. Only 14 references.
Response 4: Thank you for this comment. We have improved the whole discussion section and added more references.
5、Comment 5:strengths and limitations of this study (Page 7, lines 36), is this the first study conducted in this region?
Response 5: There was a previous study about overall blood service in the Tibetan regions of Garzê and Aba during 2013 to 2016. Our study is the first to investigate blood service including the geography, population, blood services in each counties in Aba. It will help us to know blood service in Aba in detail.
Wang, Y.; Wu, Z.; Yin, Y.H.; Rao, S.Q.; Liu, B.; Huang, X.Q.; Liu, X.X.; Li, W.H.; Ye, S.L.; Li, S.Y., et al. Blood service in the Tibetan regions of Garzê and Aba, China: a longitudinal survey. Transfus Med 2017, 27, 408-412, doi:10.1111/tme.12468.
Reviewer 2 Report (Previous Reviewer 4)
The paper is a good descriptive study of a niche problem. It is detailed and well documented. The authors have added significantly and meaningfully to the content of the earlier draft.
Minor language editing would improve the quality of the paper.
Author Response
1、Comment 1:the manuscript has many small errors of grammar and syntax. If a native English speaker is available these small errors can be easily corrected.
Response 1: Thank you for this comment. We checked the manuscript carefully, and made many revisions to improve it.
2、Comment 2: In the introduction the authors note that “The agricultural and livestock population accounts for 78.1% of the total population.” Do the authors mean that people dealing with agriculture and livestock account for 78.1% of the population? Please clarify.
Response 2: “The agricultural and livestock population” means peasants and herdsman. We rewrote the sentences (Page 1, Lines 40).
“78.1% of the total population consists of peasants and herdsmen [1,2].”
3、Comment 3:The authors mention the counties and parse data by the different counties. I think adding a map as a figure showing the counties with name, and the letter the authors use to identify each county. Also adding the altitude of the county would be useful.
Response 3: The map of Aba is showed below, which could be found online. Although we did not show the map in the manuscript, in Table 1, we listed the geography and population information of the 13 countries, including the area and altitude. What’ more, in the revised manuscript, we named the countries with their full names.
Reviewer 3 Report (Previous Reviewer 2)
The revised manuscript by Pan Sun et al. has addressed all of my suggested revisions. I suggest publication after minor errors in English syntax and grammar are corrected. Please see below.
There are still minor errors with English syntax and grammar. I suggest that the journal work closely with the author(s) to correct these minor problems.
Author Response
1、Comment 1:You could discuss why people living in high altitude may be less likely to donate.
Response 1: Thank you for this comment. We added some discussion (Page 7, Lines 37-50).
“Blood was collected mostly using blood collection vehicles on the street and greatly affected by bad natural conditions. The statistical analysis in this study also suggested that the county's blood donation rate decreases as the altitude increases. Because it’s hard to conduct recruitment and blood collection for its relatively bad weather and traffic conditions. And the main population of high-altitude places is farmers, herdsmen, Tibetans with low-education [1,2]. In this study, we found that most donors are civil servants, students, Han Chinese with higher-education. Farmers and herdsmen, which were 80% of the total population, only accounted for less than 14% of the blood donor population. Most Tibetan regions are remote and economically deprived rural areas. Due to the different language, lower-education and influence of religion, people are lack of awareness of voluntary blood donation and the knowledge of health benefits of blood donation [12-14]. Moreover, the recruitment of local Tibetan residents was difficult due to their religious beliefs, where blood is regarded as a holy gift that should be well preserved [15].”
2、Comment 2:Page 2 line 33 - e.g. where plateaus instead of that.
3、Comment 3:Page 3 line 8-9 difficult to understand.
Response 3: We rewrote this sentences (Page 3, Lines 12-15).
“The comparison between counties showed that the average blood donation rate of 1000 populations in Barkam and Wenchuan was significantly higher than other counties. Moreover, we found that the blood donation rate in Hongyuan, Rangtang and Aba with county seat altitude exceeding 3000 meters, is significantly lower than other counties.”
Reviewer 4 Report (Previous Reviewer 1)
No further comments.
Author Response
1、Comment 1:Perhaps the authors could add a little more explanatory dsicussion.
Response 1: Thank you for this comment. We have improved the whole discussion section.
1、Comment 1:The author’s data showed that the blood collection and supply was imbalance. The donation frequency was 3.4/1000 in local, which is much lower than 10/1000, but data in Fig.1 showed another result that the collection and supply amount was much close. More blood products were from other transfusion service, as the author mentioned, but they failed to calculate how many blood consumption were from outside of the local blood services.
Response 1: Thank you for this comment. We did not directly calculate how many blood consumption were from outside of the local blood services, but Figure 1 showed the specific volume of blood collection and supply in Aba every year from 2013 to 2018. For example, we could see the volume of blood collection in 2013 was 701 L, while the volume of blood supply was 785 L, the shortage was worked out with external support.
2、Comment 2: In page 3, when the author talked about Fig.2, what the mean of “there was a significant difference in blood 5donation rates among different counties (P=0.000<0.05)”? The same expression of “P=0.000<0.05” were used several times.
Response 2: We used one-way ANOVA to do multi-group comparisons for blood donation among different counties, and a p-value of less than 0.05 was considered statistically significant. The p value in this comparison was 0.000, it was less than 0.05, we used an incorrect expression “P=0.000<0.05” to express this. We corrected the expression to p<0.05.
3、Comment 3: Author stated that “The special characteristics of Aba Tibetan and Qiang Regions makes the blood donation service there different from other parts of China”, but they failed to compare the difference and listed other region’s data, especially similar region such as Tibet or Uygur AutonomousRegion of China.
Response 3: We improved the discussion. The donation rate per 1000 population in Aba was compared with Tibet, Sichuan province and national rate. And we added some information about Uygur Autonomous Region (Page 7, Lines 13-17).
“The donation rate per 1000 population of this area was 3.8 in 2018, higher than Tibet (0.5 per thousand, in 2018), the largest Tibetan area in China [6], but far lower than that of Sichuan province ( 9.3 per thousand) and China (11.1 per thousand) in 2018. While in Xinjiang Uygur Autonomous region, another large ethnic region of China, the donation rate per 1000 population was 6.3 in 2018 [7,8].”
4、Comment 4: The conclusion in the current study was confused. All the transfusion services need to adapt strategy to control the transmission of epidemic diseases and to ensure adequacy and sustainable availability of quality blood. In another word, the author did not draw any novel conclusion at all.
Response 4: Through this descriptive research, we know better about the situation and the challenges that blood service in Aba. And we added more suggestion and discussion for establishing a safe and adequate blood service system in Aba in the future.
5、Comment 5: The English writing need native speakers’ polish.
Response 5: We carefully improved the English writing of this manuscript.
This manuscript is a resubmission of an earlier submission. The following is a list of the peer review reports and author responses from that submission.
Round 1
Reviewer 1 Report
Please, respond to the comments given in the pdf document!

Reviewer 2 Report
The manuscript by Pan Sun and colleagues is a useful description of blood collection and transfusion in the China's Qinghai-Tibet Plateau. The manuscript is well written and organized and will make a useful contribution to the literature. I have the following recommendations for revisions:
· The manuscript has many small errors of grammar and syntax. If a native English speaker is available these small errors can be easily corrected
· In the introduction the authors note that “The agricultural and livestock population accounts for 78.1% of the total population.” Do the authors mean that people dealing with agriculture and livestock account for 78.1% of the population? Please clarify
· The authors mention the counties and parse data by the different counties. I think adding a map as a figure showing the counties with name, and the letter the authors use to identify each county. Also adding the altitude of the county would be useful.
Reviewer 3 Report
Interesting and important paper - may be used to further improve the blood services, in the mentioned area.
You could discuss why people living in high altitute may be less likely to donate
Page 2 line 33 - e.g.where plateaus instead of that
Page 3 line 8-9 difficult to understand
Reviewer 4 Report
This is a basic descriptive study with some explanatory power. Its value is the additional information and insights it provides about management of blood services in remote rural areas. The generalizability of its findings may be limited, as the authors state, but it may provide insights about similar issues in other special locations. Perhaps the authors could add a little more explanatory dsicussion.
Reviewer 5 Report
The current study reported that in Aba Prefecture of China, the amount of blood donation and use increased by 29% and 45% in the past 6 years. The donation rate was 3.4 ‰ and the per capital blood use was 1.04 mL. The blood donation rate in Tibetans and Qiang people is much lower compared to Han nationality. The rejection rate of blood in laboratory testing was higher than the average levels of China. The author stated that these maybe attributable to the lower education levels, bad geographical and poor medical conditions. Basically, the current report was a retrospective investigation and the main findings were not novel. Most importantly, the author failed to provide any suggestive strategy to change the situation. Furthermore, the research design was not reasonable, which made the conclusion was not reliable. The following concerns should be addressed before publication.
1. The author’s data showed that the blood collection and supply was imbalance. The donation frequency was 3.4/1000 in local, which is much lower than 10/1000, but data in Fig.1 showed another result that the collection and supply amount was much close. More blood products were from other transfusion service, as the author mentioned, but they failed to calculate how many blood consumption were from outside of the local blood services.
2. In page 3, when the author talked about Fig.2, what the mean of “there was a significant difference in blood 5donation rates among different counties (P=0.000<0.05)”? The same expression of “P=0.000<0.05” were used several times.
3. Author stated that “The special characteristics of Aba Tibetan and Qiang Regions makes the blood donation service there different from other parts of China”, but they failed to compare the difference and listed other region’s data, especially similar region such as Tibet or Uygur Autonomous Region of China.
4. The conclusion in the current study was confused. All the transfusion services need to adapt strategy to control the transmission of epidemic diseases and to ensure adequacy and sustainable availability of quality blood. In another word, the author did not draw any novel conclusion at all.
5. The English writing need native speakers’ polish.